# VIDEO-KTR: REINFORCING VIDEO REASONING VIA KEY TOKEN ATTRIBUTION

**Ziyue Wang[1,2], Sheng Jin[1], Zhongrong Zuo[1], Jiawei Wu[3],**
**Han Qiu[1], Qi She[1], Hao Zhang[4,†], Xudong Jiang[2,†]**

[1]ByteDance

[2]School of Electrical and Electronic Engineering, Nanyang Technological University

[3]National University of Singapore

[4]College of Computing and Data Science, Nanyang Technological University

`ziyue005@e.ntu.edu.sg, hzhang26@outlook.com, exdjiang@ntu.edu.sg`

## ABSTRACT

Reinforcement learning (RL) has shown strong potential for enhancing reasoning in multimodal large language models, yet existing video reasoning methods often rely on coarse sequence-level rewards or single-factor token selection, neglecting fine-grained links among visual inputs, temporal dynamics, and linguistic outputs, limiting both accuracy and interpretability. We propose **Video-KTR**, a modality-aware policy shaping framework that performs selective, token-level RL by combining three attribution signals: (1) *visual-aware* tokens identified via counterfactual masking to reveal perceptual dependence; (2) *temporal-aware* tokens detected through frame shuffling to expose temporal sensitivity; and (3) *high-entropy* tokens signaling predictive uncertainty. By reinforcing only these key tokens, Video-KTR focuses learning on semantically informative, modality-sensitive content while filtering out low-value tokens. Across five challenging benchmarks, Video-KTR achieves state-of-the-art or highly competitive results—42.7% on Video-Holmes (surpassing GPT-4o)—with consistent gains on both reasoning and general video understanding tasks. Ablation studies verify the complementary roles of the attribution signals and the robustness of targeted token-level updates. Overall, Video-KTR improves accuracy and interpretability, offering a simple, drop-in extension to RL for complex video reasoning. Our code and models are available at `https://github.com/zywang0104/Video-KTR`.

## 1 INTRODUCTION

Reinforcement learning (RL) has emerged as a powerful paradigm for enhancing long-chain reasoning in large language models (LLMs) (OpenAI, 2024; DeepSeek-AI, 2025; Yang et al., 2025). In particular, GRPO (Shao et al., 2024) exhibits strong performance on complex reasoning tasks. Building on this success, RL has been extended to multimodal LLMs (MLLMs), achieving notable gains on visual understanding through sample diversification (Wang et al., 2025a; Leng et al., 2025), tailored reward design (Tan et al., 2025; Shen et al., 2025), and advanced optimization strategies (Deng et al., 2025; Zhang et al., 2025; Dang et al., 2025). Yet, these advances primarily target static images, leaving video reasoning comparatively underexplored. Recent work mitigates this gap by integrating temporal supervision, *e.g.*, contrasting predictions on ordered and shuffled frames, to strengthen temporal sensitivity (Feng et al., 2025), and by embedding spatio-temporal priors into reward mechanisms to better align spatial grounding with temporal ordering (Li et al., 2025b; Sun et al., 2025). Other studies build RL-driven tool agents capable of decomposing and solving long-form video queries (Tian et al., 2025), as well as scalable reward designs that adapt to video complexity via difficulty-aware or temporally robust objectives (Park et al., 2025; Chen et al., 2025b).

However, video reasoning still struggles to accurately model temporal dynamics and exploit visual cues. Many approaches overlook explicit temporal dependencies and causal structures—essential

---

†Correspondence to Hao Zhang and Xudong Jiang.

for effective reasoning (Tian et al., 2025; Huang et al., 2025). Some methods (Feng et al., 2025) introduce temporal-specific constraints, such as penalizing predictions on shuffled frames, yet these rely on coarse global assumptions. This design ignores tasks solvable by static cues, introducing optimization noise that may hinder training. Moreover, many approaches fail to ensure fine-grained semantic alignment between visual inputs and output tokens. Without explicit modeling of modality-specific dependencies, models underutilize visual evidence and over-rely on linguistic priors (Wang et al., 2025d; Li et al., 2025b), increasing hallucination risks. Even token-level RL prioritizing high-uncertainty tokens via entropy (Wang et al., 2025b) often lack modality awareness, limiting their ability to capture visual–temporal dependencies. The absence of precise token-level correspondence across modalities not only reduces reasoning accuracy but also limits interpretability.

To address these challenges, we propose **Video-KTR**, a modality-aware policy shaping framework for token-level optimization in video reasoning. Unlike prior methods that impose coarse, global temporal constraints (Feng et al., 2025; Dang et al., 2025) or rely solely on entropy-based token selection (Wang et al., 2025b), Video-KTR explicitly models token sensitivity to visual and temporal perturbations, updating only critical tokens. We categorize these tokens as: (1) *visual-aware tokens*, identified via counterfactual masking to capture reliance on perceptual input; (2) *temporal-aware tokens*, detected through frame shuffling to reflect sensitivity to temporal order and causality; and (3) *high-entropy tokens*, characterized by predictive uncertainty, complementing the first two categories. This selective updating focuses learning capacity on the most informative reasoning steps, reducing interference from redundant tokens. As a result, Video-KTR significantly enhances both accuracy and integrated multimodal reasoning in complex video tasks.

To assess Video-KTR, we conduct extensive experiments on representative video reasoning benchmarks. Results demonstrate SOTA or highly competitive performance across datasets. On Video-Holmes, our method achieves an overall accuracy of $42.7\%$, surpassing GPT-4o ($42.0\%$). Moreover, ablation experiments show that modality-aware token attribution, selective reinforcement learning, and targeted perturbation jointly enhance the exploitation of visual and temporal cues. These findings confirm that Video-KTR's fine-grained, token-level optimization provides an effective and interpretable approach to advancing video reasoning.

**Our contributions** are twofold: 1) We employ counterfactual analysis to unveil modality-sensitive critical tokens in video reasoning, and propose Video-KTR tailored for token-level optimization in video reasoning tasks. To the best of our knowledge, this is the first work to integrate modality-aware token selection into RL for video reasoning. 2) Video-KTR achieves SOTA performance on multiple video reasoning benchmarks, including 42.7% on Video-Holmes dataset—matching GPT-4o and significantly outperforming existing open-source baselines. Ablation studies confirm the effectiveness of our modality-attribution signals, token selection strategy, and framework design, alongside the robustness and generalizability of Video-KTR.

## 2 RELATED WORK

Reinforcement learning has emerged as a powerful paradigm for enhancing the reasoning abilities of LLMs (OpenAI, 2024; DeepSeek-AI, 2025; Yang et al., 2025), while GRPO (Shao et al., 2024) and its variants are particularly influential in optimizing outcome-level correctness. Upon these advances, RL has been increasingly extended to multimodal LLMs (Wang et al., 2025c; Huang et al., 2025; Leng et al., 2025; Meng et al., 2025; Yuan et al., 2025; Zhang et al., 2025; Chen et al., 2025a). However, these methods remain centered on visual perception, with limited attention to temporal dynamics. Recent research proposes specialized RL strategies tailored for video reasoning, including temporally-aware fine-tuning by contrasting predictions from shuffled versus ordered frames (Feng et al., 2025), enhancing spatio-temporal reasoning via structured rewards (Li et al., 2025b; Sun et al., 2025), introducing tool-based agent frameworks for handling long-form video tasks (Tian et al., 2025), and developing scalable reward designs such as difficulty-aware regression and robust temporal reward formulations (Park et al., 2025; Chen et al., 2025b). Despite notable progress, these methods predominantly rely on coarse sequence-level rewards or limited token-level selection methods without explicit modeling of modality-specific dependencies. In contrast, we propose to jointly model complementary visual, temporal, and high-entropy token attribution signals, enabling more precise token-level credit assignment and improving reasoning performance. To the best of our

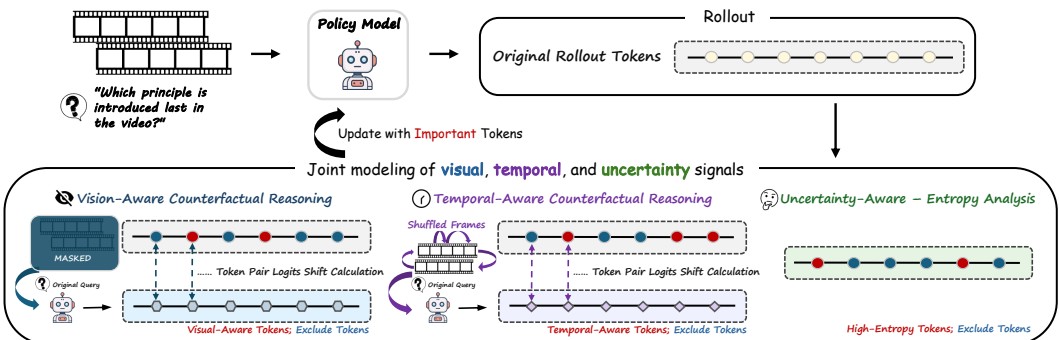

Figure 1: Overview of the Video-KTR framework. The model identifies key tokens based on entropy, visual, and temporal signals, and updates only those tokens during reinforcement learning.

knowledge, this is the first work to unify multiple token attribution strategies for modality-sensitive RL in Video-MLLMs.

## 3 METHODOLOGY

Recent RL work for Video-MLLMs (Feng et al., 2025; Chen et al., 2025b) mainly relies on coarse sequence-level rewards or simple token-level heuristics that capture only predictive uncertainty, without modeling video-specific structures like visual–temporal coherence or causal dependencies. In this work, we propose **Video-KTR**, a modality-aware policy shaping framework (Figure 1) that performs fine-grained RL through token-level attribution. We use counterfactual analysis to identify tokens conditioned on visual cues and temporal order, and exploit predictive entropy to detect reasoning-critical tokens. By selectively updating these tokens, Video-KTR amplifies reasoning-sensitive signals, improving both accuracy and interpretability.

### 3.1 MULTI-PERSPECTIVE TOKEN IMPORTANCE ANALYSIS

A central challenge in video reasoning is identifying which outputs truly rely on visual or temporal cues. Prior attribution methods, such as entropy-based approaches (Wang et al., 2025b), expose uncertain tokens but cannot distinguish modality-specific dependence from generic reasoning difficulty. To address this, we propose a modality-aware attribution framework that jointly captures visual dependence, temporal sensitivity, and predictive uncertainty, enabling precise identification of key reasoning tokens.

**Visual-Aware Tokens.** Tokens closely associated with visual input are vital for multimodal reasoning, as they align language outputs with perceptual evidence and enable key functions like reference resolution, object recognition, and appearance description. These capabilities are particularly important in tasks that involve spatial localization, appearance-based inference, and dynamic scene understanding. Effective targeted learning thus hinges on accurately identifying output components that genuinely rely on visual information.

To identify these visual-aware tokens, we introduce a visual attribution mechanism based on counterfactual perturbation, which quantifies each token's sensitivity to visual input by masking the video and measuring the resulting shift in logits. Given a video input $\mathbf{v}$ and textual prompt $\mathbf{t}$, the model generates a response sequence $\hat{y} = (y_1, y_2, \ldots, y_T)$, with decoder logits $z_i$ for each token $y_i$. We first compute logits under both full-context and masked-visual settings, extract the target token's logit at each position as:

$$\mathbf{z_i^{\text{visible}}} = f_\theta(y_i \mid y_{<i}, \mathbf{v}, \mathbf{t}), \quad \mathbf{z_i^{\text{masked}}} = f_\theta(y_i \mid y_{<i}, \tilde{\mathbf{v}}, \mathbf{t}),$$

where $f_\theta$ is the decoder network, $\mathbf{v}$ denotes the original visual input and $\tilde{\mathbf{v}} = \mathbf{0}$ indicates visual features are masked. Then, the visual attribution score is calculated by the log-probability shift as:

$$\Delta_i^{\text{vis}} = \left| \log \text{softmax}(\mathbf{z_i^{\text{full}}})_{y_i} - \log \text{softmax}(\mathbf{z_i^{\text{masked}}})_{y_i} \right|,$$

where $\Delta_i^{\text{vis}}$ assesses the visual dependence of $i$-th token. A larger $\Delta_i^{\text{vis}}$ indicates that the prediction of token $y_i$ is strongly dependent on visual input. Depicted in Figure 2a, tokens such as "person," "door," and "blue" exhibit high visual sensitivity and are prioritized during training to strengthen perceptual grounding.

**Temporal-Aware Tokens.**    Tokens sensitive to the temporal structure of videos are crucial for understanding event order, causality, and temporal dynamics. They allow the model to distinguish temporal relations, reason about progression, and track changes over time. To enhance temporal reasoning, it is essential to identify output tokens significantly affected by the sequence of visual events. Unlike T-GRPO (Feng et al., 2025), which applies sequence-level penalties to temporally shuffled inputs, we instead compute logit shifts induced by frame shuffling to locate tokens most responsive to temporal cues, enabling more fine-grained and targeted model optimization. To identify temporal-aware tokens, we disrupt the input's temporal order by shuffling the video frames, thereby breaking the original event sequence, and compute the resulting token-wise logit shifts. Given a video sequence $\mathbf{v} = (v_1, v_2, \ldots, v_F)$ and a textual prompt $\mathbf{t}$, the model generates a response sequence $\hat{y} = (y_1, y_2, \ldots, y_T)$, with decoder logits $z_i$ for each token $y_i$. Similarly, we extract the target token's logit at each position as:

$$\mathbf{z_i^{\text{ordered}}} = f_\theta(y_i \mid y_{<i}, \mathbf{v}, \mathbf{t}), \quad \mathbf{z_i^{\text{shuffled}}} = f_\theta(y_i \mid y_{<i}, \hat{\mathbf{v}}, \mathbf{t}),$$

when $f_\theta$ denotes the decoder network, and the temporal attribution score for each token is computed as:

$$\Delta_i^{\text{temp}} = \left| \log \text{softmax}(\mathbf{z_i^{\text{ordered}}})_{y_i} - \log \text{softmax}(\mathbf{z_i^{\text{shuffled}}})_{y_i} \right|.$$

Similar to visual-aware tokens, a larger $\Delta_i^{\text{temp}}$ indicates stronger reliance on the video's temporal structure. Tokens such as "first," "then," and "appear" with high $\Delta_i^{\text{temp}}$ (Figure 2b) are highly sensitive to event progression and transitions, making them essential for modeling event flow and long-range dependencies beyond isolated frames. This attribution mechanism not only pinpoints temporally sensitive reasoning cues but also highlights where temporal understanding is most critical. In video question answering, many queries depend on capturing action sequences and event relations. For instance, answering "*What did the person do after entering the room?*" requires precise temporal reasoning. Our temporal-aware attribution explicitly reveals such dependencies, guiding reinforcement learning toward causally significant moments.

**Entropy-Aware Tokens.**    While visual and temporal attributions capture modality-specific dependencies, they may miss reasoning-critical tokens unrelated to perception or sequence. Thus, we incorporate predictive entropy (Wang et al., 2025b) as a logic-aware criterion to identify low-confidence tokens:

$$\mathcal{H}(i) = -\sum_v p(z_i = w) \log p(z_i = w),$$

where $w$ denotes token index. High-entropy tokens—*e.g.*, "however," "wait"—often mark discourse pivots or decision points critical for reasoning. Illustrated in Figure 2c, they capture semantic ambiguity, conflicting cues, or implicit causality, and offer a logic-aware means of identifying reasoning signals beyond visual and temporal cues. We incorporate it into token-level credit assignment as an empirically validated strategy.

## 3.2   Token Selection and Policy Update

Building on the token importance estimation, we explore how to integrate this information into the RL process effectively. The central challenge lies in *how to leverage the identified key tokens to enhance the training efficiency*. Conventional RL frameworks like GRPO treat all tokens equally during reward assignment, which dilutes learning signals—particularly in long video reasoning sequences—and thus hampers training efficiency. We introduce a **token-based policy shaping** mechanism that selectively reinforces semantically critical tokens during training. To be specific, we select the top $r\%$ of tokens from each attribution strategy and take their union, $\mathcal{S} = \mathcal{S}_{\text{vis}} \cup \mathcal{S}_{\text{temp}} \cup \mathcal{S}_{\text{ent}}$, as the set of **key reasoning tokens**. These tokens collectively capture the most essential semantic cues for visual grounding, temporal understanding, and decision uncertainty, thereby enabling more focused and efficient policy updates.

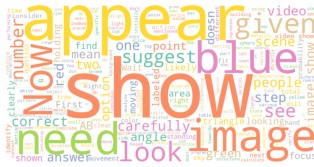
(a) Visual-aware tokens.

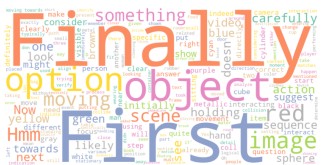
(b) Temporal-aware tokens.

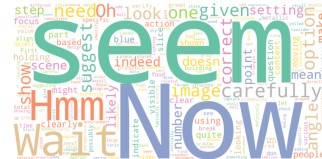
(c) Entropy-aware tokens.

Figure 2: An illustration of critical tokens identified by each attribution strategy.

We adopt the GRPO framework (Shao et al., 2024), which improves the current policy $\pi_\theta$ by contrasting its outputs with a set of reference completions sampled from the previous policy $\pi_{\theta_{old}}$. Given a batch of $G$ questions $q \sim P(Q)$ and corresponding rollouts $o_{i\,i=1}^{G} \sim \pi_{\theta_{old}}(\cdot|q)$, the GRPO objective is defined as:

$$\mathcal{J}_{\text{GRPO}}(\theta) = \mathbb{E}_{q,\{o_i\}}\left[\frac{1}{G}\sum_{i=1}^{G}\frac{1}{|o_i|}\sum_{t=1}^{|o_i|}\min\left(r_{i,t}\cdot\hat{A}_{i,t}, \text{clip}(r_{i,t}, 1-\epsilon, 1+\epsilon)\cdot\hat{A}_{i,t}\right)\right], \quad (1)$$

where the KL divergence term is omitted for simplicity, $r_{i,t} = \frac{\pi_\theta(o_{i,t}|q,o_{i,<t})}{\pi_{\theta_{old}}(o_{i,t}|q,o_{i,<t})}$ is the token-level likelihood ratio, and $\hat{A}_{i,t}$ denotes the advantage estimate computed via the normalized group reward for all tokens in $o_i$. To prioritize semantically important reasoning tokens during learning, we apply a binary mask $m_{i,t} \in \{0, 1\}$ to selectively update tokens. This mask is constructed from the union of top-ranked tokens identified by visual-aware, temporal-aware, and entropy-aware attributions. The resulting objective is:

$$\mathcal{J}_{\text{Video-KTR}}(\theta) = \mathbb{E}_{q,\{o_i\}}\left[\frac{1}{G}\sum_{i=1}^{G}\frac{1}{|o_i|}\sum_{t=1}^{|o_i|}m_{i,t}\cdot\min\left(r_{i,t}\cdot\hat{A}_{i,t}, \text{clip}(r_{i,t}, 1-\epsilon, 1+\epsilon)\cdot\hat{A}_{i,t}\right)\right], \quad (2)$$

where only tokens with $m_{i,t} = 1$, *i.e.*, those identified as important, contribute to the loss, allowing the model to focus supervision on semantically rich segments such as temporally anchored verbs, visually descriptive nouns, and discourse pivots with high uncertainty. As shown in Figure 2, the selected tokens differ across attribution strategies, reflecting their distinct roles in multimodal reasoning. In contrast, unselected tokens are largely low-information function words (*e.g.*, auxiliary verbs, pronouns, determiners, prepositions, and punctuation), as illustrated in Figure 6b. Their prevalence further validates the attribution mechanism's ability to filter redundant content and enhance learning efficiency.

## 4 EXPERIMENTS

### 4.1 EXPERIMENTAL SETUP

**Datasets.** For RL data, we select the raw data released by Video-R1 (Feng et al., 2025), containing 260K samples with verifiable answers. We conduct strict difficulty sampling to enhance the overall difficulty and coverage of the training samples. Specifically, we remove samples that achieve above 80% or below 20% accuracy over 8 tails using Video-R1-SFT (Feng et al., 2025), leading to $\sim 15K$ high-quality subset. We also incorporate 1.5K training examples of Video-Holmes (Cheng et al., 2025) to form the final training set.

**Implementations.** The Video-KTR directly perform RL training on the Video-R1 SFT checkpoint, which itself is built upon the Qwen-2.5-VL-7B (Bai et al., 2025) backbone. To improve the efficiency, we limit the input to a maximum of 16 frames with a maximum resolution of $128\times28\times28$ pixels. We set the learning rate to $2e-6$, the global batch size to 32, the rollout batch size to 256, the maximum sequence length to 16384; the KL divergence coefficient ($\beta$) is configured at $0.4$; and 8 responses are sampled for each prompt with a temperature of $1.0$. During the inference, we may extend the maximum frames to 64 with a maximum resolution of $256 \times 28 \times 28$.

Table 1: Performance comparison across reasoning-oriented and general-purpose video benchmarks, where * indicates scores evaluated by ourselves.

| Models | Size | # Frames | Video Reasoning Benchmark | | | Video General Benchmark | |
|---|---|---|---|---|---|---|---|
| | | | Video-Holmes | VideoMMMU | MMVU(mc) | TempCompass | VideoMME |
| **Proprietary MLLMs** | | | | | | | |
| GPT-4o | – | – | 42.0 | 61.2 | 75.4 | 73.8 | 71.9 |
| GPT-5 | – | – | 46.7* | 84.6 | 82.6* | 83.3* | 86.7 |
| Gemini-1.5-Pro | – | – | 41.3 | 53.4 | 71.2 | 67.1 | 75.0 |
| Gemini-2.5-Pro | – | – | 45.0 | 83.6 | 78.4* | 84.3* | 84.3 |
| **Open-Source MLLMs** | | | | | | | |
| LLaVA-OV | 7B | 64 | – | 33.8 | 49.2 | 64.2 | 58.2 |
| VILA-1.5 | 8B | 64 | – | 33.8 | 49.2 | 58.8 | 58.2 |
| Qwen2.5-VL | 7B | – | 27.8 | 47.4 | 59.2 | 67.9 | **65.1** |
| Video-R1 | 7B | 32 | 36.5 | 52.3 | 63.8 | 73.2 | 59.3 |
| Video-RTS | 7B | 51.2 | 40.7 | 52.7 | 66.4 | – | 63.0 |
| TW-GRPO | 7B | 16 | 32.9 | 51.3 | 65.8 | 73.3 | 55.1 |
| **SFT Models** | | | | | | | |
| Qwen2.5-VL-SFT | 7B | 16 | 31.7 | 47.4 | 61.3 | 69.2 | 52.8 |
| | | 32 | 33.9 | 49.4 | 63.5 | 69.9 | 55.4 |
| | | 64 | 33.7 | 49.4 | 61.6 | 70.0 | 58.8 |
| **Video-KTR** | 7B | 16 | 40.7 | 51.3 | 65.7 | 73.3 | 57.3 |
| | | 32 | 41.6 | 52.6 | 65.9 | 73.4 | 60.3 |
| | | 64 | **42.7** | **53.1** | **66.6** | **73.5** | 62.5 |

**Benchmarks.** To comprehensively assess the performance of Video-KTR, we select five challenging video benchmarks: (1) reasoning-oriented benchmarks, including **Video-Holmes** (Cheng et al., 2025), **VideoMMMU** (Hu et al., 2025), and **MMVU** (Zhao et al., 2025), to assess the temporal, causal, and knowledge-intensive reasoning capability of MLLMs; (2) general understanding benchmarks, including **TempCompass** (Liu et al., 2024) and **VideoMME** (Fu et al., 2025). Note Video-Holmes is a newly established benchmark that poses significant challenges, requiring models to integrate dispersed narrative cues from suspenseful short films and perform complex deductive reasoning beyond basic visual recognition.

**Baselines.** We compare Video-KTR against a diverse set of baselines, encompassing both general-purpose and reasoning-oriented MLLMs of similar size: LLaVA-OneVision (LLaVA-OV; Li et al. (2025a)), VILA-1.5 (Liu et al., 2025), Qwen2.5-VL (Bai et al., 2025), Video-R1 (Feng et al., 2025), Video-RTS (Wang et al., 2025d), and TW-GRPO (Dang et al., 2025).

## 4.2 RESULTS AND ANALYSIS

**Comparison with state-of-the-art models.** We evaluate Video-KTR against a diverse set of video reasoning models on three reasoning and two general video understanding benchmarks. Reported in Table 1, Video-KTR consistently achieves state-of-the-art results. On **Video-Holmes**—which targets high-level temporal and social reasoning in short films—our model attains **42.7%**, surpassing all open-source baselines and closely matching proprietary models like GPT-5 (OpenAI (2025); 46.7%) and Gemini-2.5-Pro (Comanici et al. (2025); 45.0%). On knowledge-intensive benchmarks, *i.e.*, **MMVU(mc)** and **VideoMMMU**, Video-KTR achieves **66.6%** and **53.1%**, respectively, highlighting its strength in complex reasoning. Video-KTR also delivers competitive performance on general video understanding, showing that improved reasoning does not compromise broad comprehension. Notably, accuracy increases steadily with more input frames (16 to 64), underscoring the scalability and robustness of our token-aware RL for longer temporal sequences.

**Research Question (RQ1): How effective is Video-KTR compared with other video RL methods?** To isolate the effect of training data and recipe, we conduct a controlled study where all

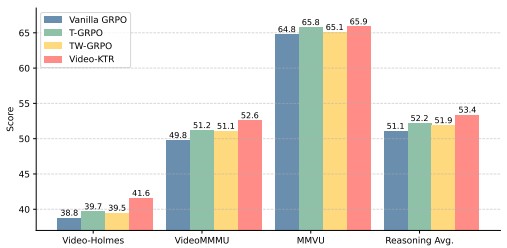
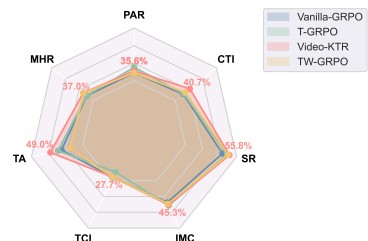

(a) Cross-Benchmark Comparison.

(b) Performance across Video-Holmes subtasks.

Figure 3: Performance comparison on (a) video reasoning benchmarks and (b) detailed subtasks of the Video-Holmes benchmark. Subtask abbreviations: **SR** = *Social Reasoning*, **IMC** = *Intention & Motive Chaining*, **TCI** = *Temporal Causal Inference*, **TA** = *Timeline Analysis*, **MHR** = *Multimodal Hint Reasoning*, **PAR** = *Physical Anomaly Reasoning*, **CTI** = *Core Theme Inference*.

Table 2: Performance of token selection using entropy (E), visual (V), and temporal (T) attributions.

| Strategy | E | V | T | Video-Holmes | Video-MMMU | MMVU | Avg |
|---|---|---|---|---|---|---|---|
| Vanilla GRPO | ✗ | ✗ | ✗ | 38.8 | 49.8 | 64.8 | 51.1 |
| E | ✓ | ✗ | ✗ | 39.5 | 49.2 | 64.3 | 51.0 |
| V | ✗ | ✓ | ✗ | 40.5 | 51.9 | 65.1 | 52.5 |
| T | ✗ | ✗ | ✓ | **42.1** | 50.1 | 65.5 | 52.6 |
| V & T | ✗ | ✓ | ✓ | 41.3 | 52.1 | 65.2 | 52.9 |
| E & T | ✓ | ✗ | ✓ | 39.6 | 50.9 | 64.3 | 51.6 |
| V & E | ✓ | ✓ | ✗ | 41.0 | 51.1 | 66.4 | 52.8 |
| V & E & T | ✓ | ✓ | ✓ | 41.6 | **52.6** | **65.9** | **53.4** |

models are trained under an identical dataset and training recipe. Depicted in Figure 3a, Video-KTR consistently outperforms vanilla GRPO, T-GRPO, and TW-GRPO, demonstrating its effectiveness independent of data quality or quantity. Figure 3b reports Video-Holmes results by subtask. Video-KTR achieves marked gains in Timeline Analysis (**TA**) and Core Theme Inference (**CTI**), tasks that require advanced temporal reasoning and holistic content understanding. The results confirm that our method enhances both temporal and structural reasoning, validating the token-aware training design. A qualitative comparison with T-GRPO (Appendix B.6) illustrates that Video-KTR identifies key visual and temporal cues more effectively, leading to correct predictions and highlighting the benefit of token-level optimization in aligning model attention with critical signals.

**RQ2: What is the impact of different attribution signals on video reasoning performance?** To assess the impact of different attribution signals, we ablate the three token types—entropy-aware (E), visual-aware (V), and temporal-aware (T)—as shown in Table 2. Each signal individually outperforms vanilla GRPO, with temporal-aware tokens giving the largest gain on Video-Holmes, reflecting their importance for sequence reasoning. However, using temporal-aware tokens alone hurts performance on other benchmarks (e.g., Video-MMMU), suggesting that a single signal can bias the policy toward a narrow reasoning mode. Pairwise combinations yield further improvements, and the full E+V+T setup consistently achieves the best results across benchmarks. These findings show that the three signals are complementary and that jointly leveraging them enables more accurate and robust policy optimization.

**RQ3: What are the linguistic divergences in token selection across attribution strategies?** To assess the complementarity of our attribution strategies, we analyze the linguistic distribution and overlap of selected tokens. As shown in Figure 4, visual-aware tokens are mostly NOUNs (24.8%), consistent with their role in object grounding. Temporal-aware tokens emphasize VERBs (21.2%) and PRONs (11.0%), reflecting action transitions and temporal references. Entropy-aware tokens contain more ADVs (8.8%), capturing discourse modulation and uncertainty. These patterns suggest that modality-aware token selection leverages complementary reasoning dimensions—perceptual grounding, temporal structure, and uncertainty cues—to enhance video understanding.

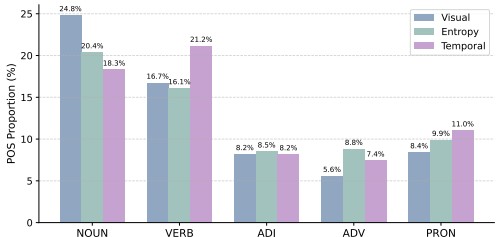
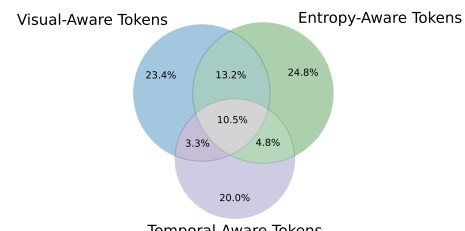

Figure 4: POS ratio across selection strategies.

Figure 5: Distribution of updated tokens.

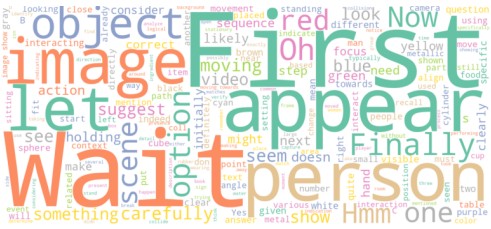
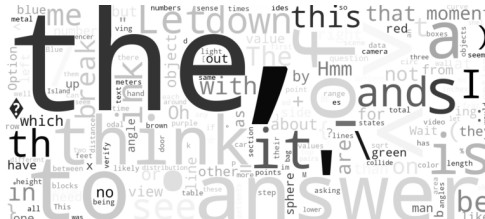

(a) Word cloud of selected critical tokens.

(b) Word cloud of unselected low-value tokens.

Figure 6: Comparison of tokens selected as reasoning-critical and unselected tokens. Selected tokens concentrate on semantically informative content, whereas unselected tokens are dominated by function words, indicating that the selection mechanism emphasizes reasoning-critical tokens while filtering low-value ones.

As illustrated in Figure 6a, the word cloud of selected tokens highlights semantically rich cues that directly contribute to multimodal reasoning, such as content words related to objects (*object*, *person*), actions (*appear*, *hold*), and reasoning markers (*wait*, *let*, *consider*, *finally*). These tokens capture critical visual and temporal evidence, guiding the model to focus on informative regions of the input. In contrast, as shown in Figure 6b, unselected tokens are dominated by low-information function words, including auxiliary verbs (AUX: *is*, *have*), pronouns (PRON: *I*, *we*), determiners (DET: *the*, *this*), prepositions (ADP: *in*, *on*), and various punctuation or formatting symbols (*e.g.*, ., ,). Their prevalence confirms the effectiveness of our attribution strategies in filtering out syntactic noise while prioritizing semantically salient reasoning cues.

**RQ4: How do token weighting strategies affect selective RL?** We compare several token-weighting strategies to examine whether soft importance modulation can improve upon the hard selection used in Video-KTR. The methods include: (1) **Binary Top-20%**, which updates only the top 20% key tokens and masks the rest; (2) **Softmax Weighting**, which normalizes importance scores into a probability distribution; (3) **Sigmoid Weighting**, which applies a smooth nonlinear mapping; (4) **Linear Weighting**, which scales updates proportionally to token scores; and (5) **Exponential Weighting**, which accentuates score gaps. As shown in Figure 7, the hard top-20% strategy consistently outperforms all soft weighting variants, indicating that strict selection yields clearer credit assignment while soft modulation spreads updates over less relevant tokens. Token-level overlap analysis (Figure 5) further shows that, after applying the union of visual, temporal, and entropy signals, the final update ratio reaches about 40%, with most tokens unique to individual strategies. This demonstrates that each attribution signal captures a distinct and complementary subset of reasoning-relevant tokens.

**RQ5: Do distributional shifts improve token attribution beyond log-probability?** To identify informative tokens for reinforcement updates, we assess both changes in predicted log-probabilities and shifts in the full output-logit distribution under modality-specific perturbations. Specifically, we quantify token-wise distances between pre- and post-perturbation logits using metrics like L1/L2 norms, KL/JS divergences, cosine similarity, and Hellinger distance. Depicted in Figure 8, the simple log-probability difference consistently outperforms these alternatives. It is robust, easy to

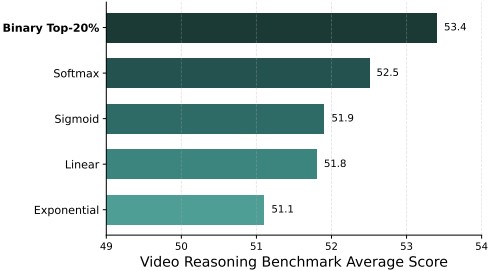

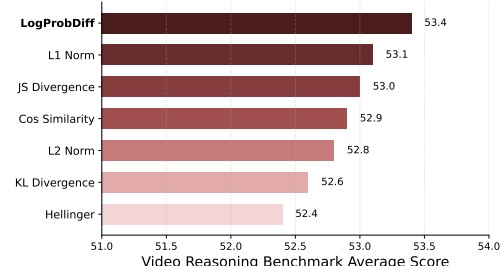

Figure 7: Benchmark Scores under Different Token Weighting Methods.

Figure 8: Benchmark Scores under Different Distance Calculation Methods.

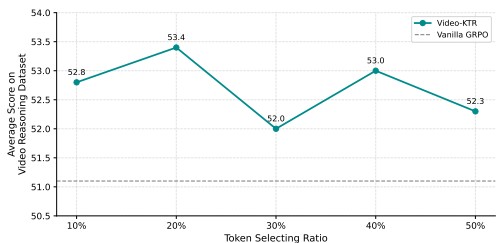

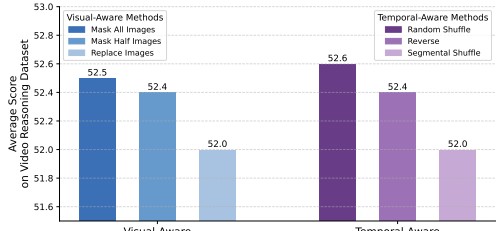

Figure 9: Effect of update ratios.

Figure 10: Effect of perturbation strategies.

implement, and efficient-requiring only a subtraction of two log-probabilities per token-making it highly scalable. These results suggest log-probability differences as a reliable and efficient signal for tracking prediction-confidence shifts and identifying informative tokens.

**RQ6: How does the token-level update ratio affect performance?** We ablate the token-level update ratio—the proportion of key tokens receiving full gradient updates—with all token types applied jointly. As shown in Figure 9, accuracy follows a double-peak pattern and achieves the best performance at a 20% ratio. Performance drops at higher ratios (30–50%), indicating that moderate selective updating is optimal, while overly large update sets introduce noise and reduce stability.

**RQ7: Do perturbation strategies influence token identification and model performance?** To assess the robustness of our token identification method, we examine alternative perturbation strategies for visual and temporal dependencies. For visual perturbation, we consider *masking all frames* (default), *masking half* , and *replacing frames with unrelated content*. For temporal analysis, we compare *random shuffling* (default), *sequence reversal*, and *segmental shuffling*. In all cases, the token update ratio is fixed at 20%. Depicted in Figure 10, all variants achieve comparable performance, with the default settings—full-frame masking (52.5) and random shuffling (52.6)—performing slightly better. These results suggest that perturbation strength has limited impact, confirming the robustness of our method. We therefore adopt the default strategies for their simplicity and reliability, without requiring manual tuning.

## 5 CONCLUSION

We introduce Video-KTR, a modality-aware policy shaping framework that enhances video reasoning by shifting RL from coarse sequence supervision to selective, token-level attribution signal assignment. By integrating visual, temporal, and uncertainty-based attribution, Video-KTR focuses on semantically critical tokens, improving both accuracy and interpretability. Experiments across five benchmarks—highlighted by 42.7% on Video-Holmes, surpassing GPT-4o—demonstrate SOTA or highly competitive performance. Ablations confirm the complementary roles of three signals and robustness of targeted updates. The approach integrates seamlessly into RL and enhances reasoning without compromising general video understanding.

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

## A    LIMITATIONS

While Video-KTR has demonstrated notable improvements and potential, several avenues remain open for further exploration. First, under challenging conditions such as low light, occlusion, rapid motion, or OCR/ASR noise, visual–text alignment may be affected. Enhancing the stability and robustness of key token selection in diverse scenarios represents a promising direction for future work. Second, Our experiments have primarily focused on video reasoning benchmarks, whereas broader applications such as video captioning or temporal grounding have not yet been systematically examined. Extending validation to these domains would offer valuable insights into the generalization of the framework. Third, the current framework emphasizes vision–language modeling without explicitly incorporating additional modalities such as audio or motion sensor data. Since these signals often provide complementary information, integrating richer multimodal inputs holds considerable potential for future research.

## B    MORE RESULTS AND ANALYSIS

### B.1    GRADIENT DECOMPOSITION AND NOISE INTERPRETATION

To analyze whether masked tokens meaningfully affect optimization, we examine gradients at the final layer, where updates most directly influence policy learning. Let the sequence-level loss be $L = \sum_t \ell_t$ and denote the final-layer gradient for token $t$ by

$$g_t = \nabla_{\theta^{(L)}} \ell_t = \frac{\partial \ell_t}{\partial h_t^{(L)}} \cdot \frac{\partial h_t^{(L)}}{\partial \theta^{(L)}},$$

where $h_t^{(L)}$ is the last-layer hidden state and $\theta^{(L)}$ denotes all parameters of the final layer. We partition tokens into a critical set $\mathcal{S}$ and a residual set (unselected low-value tokens) $\mathcal{R}$, yielding the following gradients

$$g_{\text{full}} = \sum_t g_t, \qquad g_{\text{KTR}} = \sum_{t \in \mathcal{S}} g_t, \qquad g_{\text{rest}} = \sum_{t \in \mathcal{R}} g_t.$$

To assess how these components contribute to parameter updates, we compare the directional agreement between each aggregated gradient and the full gradient. Rather than considering only their magnitudes, we evaluate whether two gradients point in similar directions in parameter space. This is measured using cosine similarity, written as $\cos(g_a, g_b)$, which reflects how much one gradient aligns with the other: values close to 1 indicate that $g_a$ and $g_b$ point in nearly the same direction and thus provide mutually reinforcing updates. In practice, we compute $\cos(g_{\text{KTR}}, g_{\text{full}})$ and $\cos(g_{\text{rest}}, g_{\text{full}})$ to quantify how much each component contributes to the effective update direction.

As shown in Figure 11, Video-KTR produces larger, cleaner, and more directionally consistent gradient updates compared to vanilla GRPO. The mean gradient norm of the selected critical tokens is 4.50, which is roughly three times higher than that of the masked tokens. Likewise, the directional

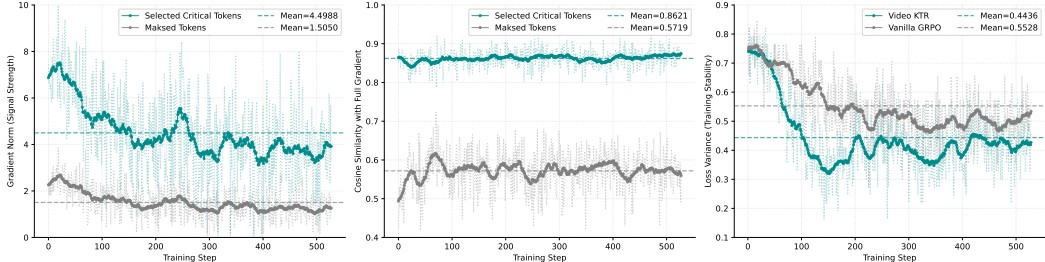

Figure 11: Video-KTR exhibits lower loss variance, stronger gradient signals, and higher gradient alignment during training. Due to the high variance of the accuracy curves, we smooth the curves using window smoothing with a window size of 20.

Table 3: Performance comparison on image benchmarks. Results show that restricting updates to visual tokens already yields clear gains over GRPO variants, highlighting the importance of visual token optimization. Nevertheless, the full Video-KTR model achieves the best performance across all benchmarks, underscoring the additional benefits of integrating temporal and entropy-aware signals.

|  | MMMU(val) | AI2D | MMSTAR | ChartQA |
|---|---|---|---|---|
| Qwen2.5-VL-SFT | 42.89 | 75.68 | 53.07 | 69.92 |
| Vanilla GRPO | 44.33 | 76.30 | 53.60 | 73.08 |
| T-GRPO | 44.67 | 77.51 | 54.32 | 77.04 |
| KTR-Visual Tokens Only | 46.44 | 79.27 | 57.38 | 77.08 |
| Video-KTR | 47.33 | 80.34 | 57.77 | 80.68 |

agreement with the full gradient is substantially stronger: the cosine similarity of critical-token gradients averages 0.862, compared with 0.572 for the masked-token gradients. These results indicate that the selected tokens dominate the effective update magnitude and align far more closely with the model's underlying optimization trajectory. In contrast, gradients from residual masked tokens are both much weaker and directionally inconsistent, suggesting that they contribute little meaningful learning signal and instead introduce noisy perturbations into the optimization process.

We further compare the training loss variance, which reflects overall optimization stability. Video-KTR achieves more stable training dynamics: its mean loss variance is 0.4436, lower than that of vanilla GRPO (0.5528) across training steps, supporting that masking low-information tokens reduces gradient noise and leads to more stable reinforcement learning updates.

Together, these results validate the effectiveness of our token-selection mechanism and highlight its role in guiding the model toward more meaningful and robust multimodal reasoning behavior.

## B.2 ADDITIONAL EVALUATION ON IMAGE BENCHMARKS

To further assess the contribution of visual tokens, we introduce a variant of our method, denoted as KTR-Visual Tokens-Only, where reinforcement learning updates are restricted to visual-related tokens while discarding temporal and high-entropy token signals. This setting isolates the effect of visual supervision and provides a controlled comparison against the full Video-KTR framework. As shown in Table 3, this variant yields notable gains over baseline GRPO methods on several classic image understanding benchmarks, including MMMU-val (Yue et al., 2024), AI2D (Kembhavi et al., 2016), MMStar (Chen et al., 2024), and ChartQA (Masry et al., 2022), underscoring the importance of visual token optimization. Nevertheless, the complete Video-KTR achieves the best overall performance, suggesting that the integration of temporal and uncertainty-aware token signals further enhances cross-modal reasoning.

Table 4: Comparison of training cost and forward efficiency. TT = training time, Mem = peak memory, SPS = samples per second, Lat. = latency. Video-KTR introduces only marginal overhead compared with Vanilla GRPO and T-GRPO.

|              | TT (h) | Mem (GB) | SPS   | fwd Lat. (s) | fwd FLOPs per GPU(T) |
|--------------|--------|----------|-------|--------------|----------------------|
| Vanilla GRPO | 4.7    | 77.9     | 0.946 | 4.70         | 116.6                |
| T-GRPO       | 5.1    | 78.3     | 0.872 | 4.89         | 128.6                |
| Video-KTR    | 5.2    | 78.5     | 0.855 | 4.92         | 129.9                |

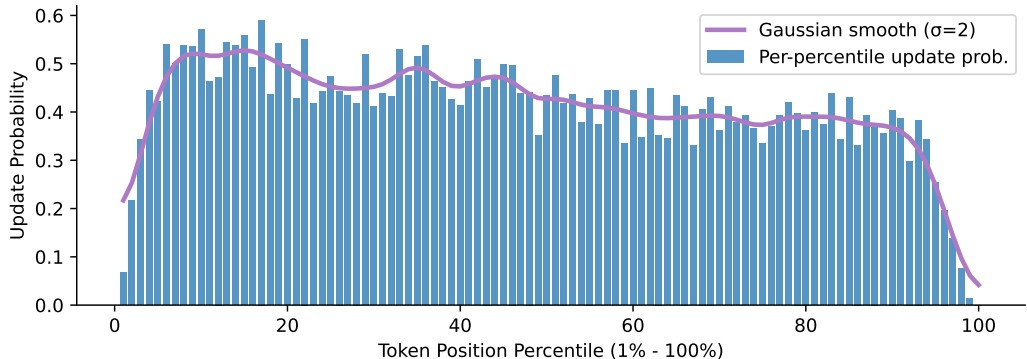

Figure 12: Distribution of updated token positions in Video-KTR. Early tokens show higher update probabilities, while later tokens still receive non-negligible updates, indicating that the model continues refining its reasoning rather than relying solely on pretrained linguistic priors.

### B.3 COMPUTATIONAL COST ANALYSIS

Our experiments were conducted on 32 H100 (80GB) GPUs, and the training of Video-KTR was completed in 5.2 hours. As summarized in Table 4, the additional computational overhead introduced by our method remains modest. The key reason is that token importance scores are computed in a single extra forward pass, avoiding repeated per-token calculations. Compared with vanilla GRPO and T-GRPO, Video-KTR shows almost no increase in peak memory usage (77.9 GB $\rightarrow$ 78.5 GB) and only a minor rise in forward latency (4.70 s $\rightarrow$ 4.92 s). The overall training time grows slightly from 4.7–5.1 hours to 5.2 hours, while forward FLOPs per GPU increase moderately (116.6 T $\rightarrow$ 129.9 T). These results indicate that our token-level reinforcement strategy delivers significant performance improvements at a marginal computational cost.

### B.4 UPDATED TOKEN POSITION ANALYSIS

To better understand where reinforcement learning updates are applied, we analyze the distribution of updated token positions. As shown in Figure 12, tokens at earlier positions exhibit consistently higher update probabilities than those at later positions. This pattern is expected: early tokens play a larger role in determining the overall direction of the answer, whereas later outputs tend to be more deterministic and less influenced by input variability. Nevertheless, we also observe non-negligible updates in later positions, suggesting that the model does not merely rely on pretrained linguistic priors but continues to refine its reasoning based on the input content.

### B.5 GENERALIZE VIDEO-KTR TO SMALLER MLLM

To assess generalizability beyond the 7B-scale backbone, we apply our training strategy to the smaller Qwen2.5-VL-3B model. As shown in Table 5, our method consistently outperforms vanilla GRPO across all frame settings (16, 32, and 64) and benchmarks. These findings demonstrate that our token-aware reinforcement learning strategy remains effective for smaller-scale MLLMs.

Table 5: Validation on Qwen2.5-VL-3B with different frame settings (16/32/64). Video-KTR consistently outperforms Vanilla GRPO, demonstrating robust improvements across benchmarks.

| Models | Frames | Video-Holmes | VideoMMMU | MMVU | TempCompass | VideoMME |
|--------|--------|--------------|-----------|------|-------------|----------|
| Vanilla GRPO | 16 | 32 | 44.0 | 58.4 | 65.2 | 51.8 |
| Video-KTR | 16 | 33.2 | 45.1 | 60.0 | 67.2 | 53.0 |
| Vanilla GRPO | 32 | 33.1 | 44.7 | 59.5 | 66.7 | 54.1 |
| Video-KTR | 32 | 36.2 | 45.3 | 61.0 | 67.9 | 56.2 |
| Vanilla GRPO | 64 | 34.1 | 45.0 | 60.3 | 66.8 | 56.0 |
| Video-KTR | 64 | 36.3 | 46.8 | 61.3 | 68.0 | 57.5 |

### B.6 QUALITATIVE PERFORMANCE OF VIDEO-KTR

To qualitatively demonstrate Video-KTR's advantages, several representative examples from Video-Holmes are presented. In Figure 13, our model correctly identifies her ability to "ignore spatial physical limitations"—the official answer—by attending to essential visual and temporal cues, such as her "disappearance and reappearance" and "unpredictable spatial movements." Conversely, T-GRPO inaccurately attributes the rule to "transformation and disguise," underscoring its inability to ground reasoning effectively in the video's actual content. This example illustrates the effectiveness of our token-level optimization in aligning model attention with critical visual and temporal signals. In Figure 14, Video-KTR further demonstrates its robustness by correctly inferring the underlying character relationship—that the two individuals are friends rather than brothers—whereas T-GRPO, lacking any supporting evidence, incorrectly selects a fraternal relationship. In Figure 15, Video-KTR accurately reasons about the character's motivation, identifying that the man pretends to be dead to escape the woman's attempted murder. In contrast, T-GRPO misidentifies the yellow gloves as a banana, leading it to the wrong conclusion that the man was awakened by the woman's phone call. In Figure 16, Video-KTR successfully tracks the temporal progression in a chaotic chase scene, recognizing that the man first tries to retreat into the room, fails, and is then relentlessly pursued. T-GRPO, however, mistakenly assumes that the man is chased continuously before attempting to return to the room, thereby misrepresenting the actual event order. These examples demonstrate Video-KTR's enhanced capacity for temporal modeling and consistent logical reasoning, allowing it to capture complex event structures with greater reliability.

## C LLM USAGE STATEMENT

In accordance with the ICLR 2026 policy on large language model (LLM) usage, we disclose that an LLM (OpenAI's ChatGPT) was used in a **limited and non-substantive capacity**. Specifically, it assisted with *minor language editing* (*e.g.*, grammar and phrasing), provided suggestions on figure color choices, and offered LaTeX formatting guidance. **No research ideas, experimental designs, analyses, or substantive scientific writing were generated by the LLM**. All scientific contributions and intellectual content are entirely the work of the authors.

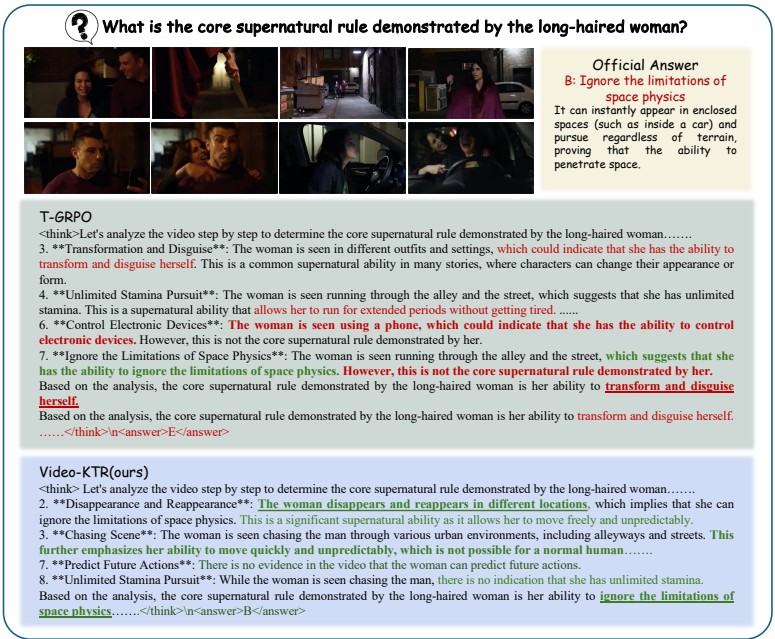

Figure 13: Qualitative comparison on Video-Holmes-Physical Anomaly Reasoning. Video-KTR infers the supernatural rule from temporal and visual cues, while T-GRPO produces an incorrect path.

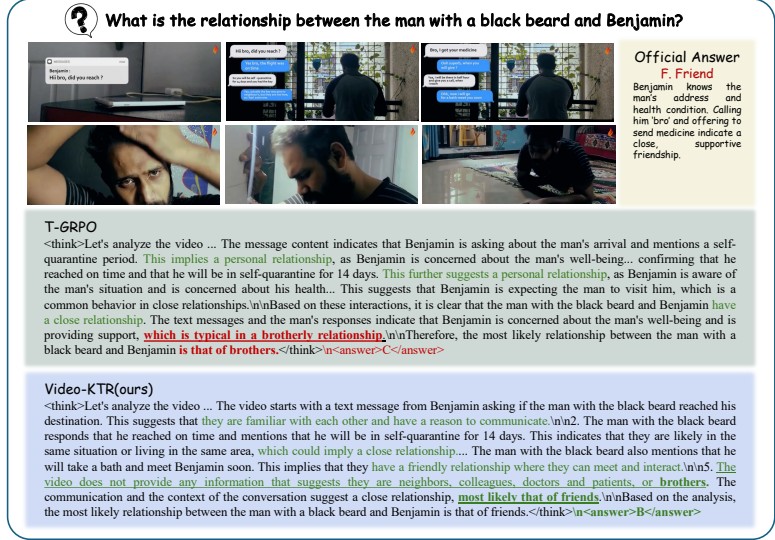

Figure 14: Qualitative comparison on Video-Holmes-Social Reasoning. Video-KTR correctly inferred the underlying character relationship, whereas T-GRPO chose an answer without factual grounding.

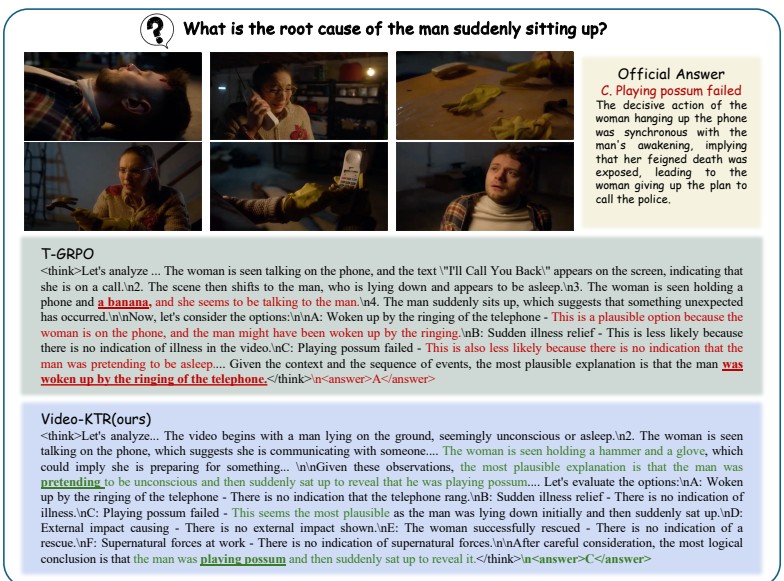

Figure 15: Qualitative comparison on Video-Holmes-Temporal Causal Inference. Video-KTR correctly reasoned about the character's motivation, whereas T-GRPO misidentified the yellow gloves as a banana and produced an incorrect answer.

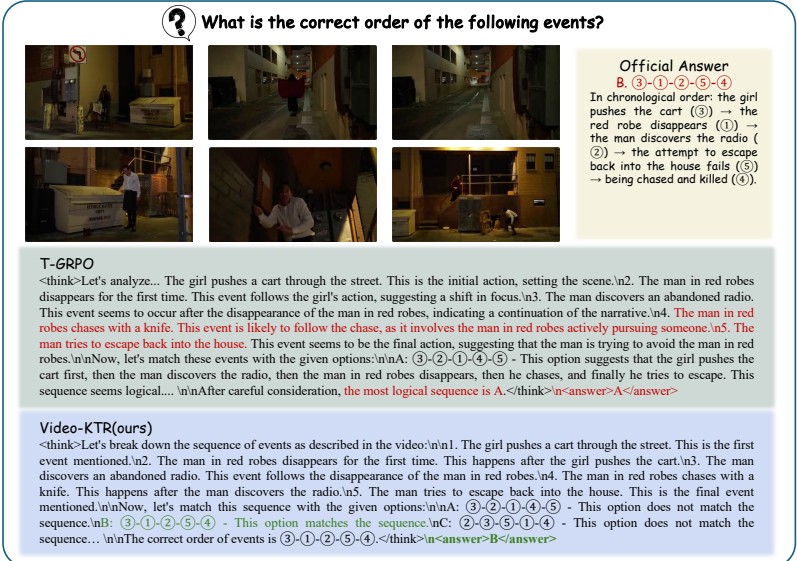

Figure 16: Qualitative comparison on Video-Holmes-Timeline Analysis. Video-KTR correctly identified the sequence of events, whereas T-GRPO confused the temporal order during the chaotic chase.

