# OpenReview forum: "Video-KTR: Reinforcing Video Reasoning via Key Token Attribution"
_ICLR.cc/2026/Conference — ICLR 2026 Poster_

### Official Review · Reviewer_UDPQ · 2025-10-17

**Soundness:** 2
**Presentation:** 3
**Contribution:** 3
**Rating:** 6
**Confidence:** 3

**Summary:**

(I tend to write shorter reviews and the length of the review does not reflect the quality of paper or the time spend on reviewing it).

Note that I am not an expert in Video modelling nor Reinforcement Learning and the review is from a perspective of a general Deep Learning researcher.

This paper proposes a few heuristics to improve the RL training of reasoning models (especially in the context of Video) by using things like perceptual dependence, remporal sensitivity and entropy based uncertainity. The authors show that each of these heuristics reveal natural points to not have RL be updating all the tokens in the trajectory but only a handful. They paper's experiments are strong and have extensive analysis. The writing is very easy to follow.

**Strengths:**

Clear explainations and good experiments. The extent of empirical investigations is commendable -- I am not an expert so can not comment on the strength of the experiments.

**Weaknesses:**

1) The comparision to the closed source models is from almost 2 generations ago and would be good to have latest to have a better understanding of where things stand.

2) While the heursitics are working great for this task at hand, given the past of DL and my experience, the heuristics stop helping in general purpose cases and while we scale up. It is a strong result for now in a narrow domain but a broader investigation is what will concretize the proposed things in the modern RL pipelines. I strongly suggest either doing a more general purpose evaluation to support this broadly.

**Questions:**

See above. Again I am not an expert so will defer to other reviewers on this paper.

---

> ### Author Response · Authors · 2025-11-24
>
> We thank the reviewer for raising these important points.
>
> ### 1. Comparison with the Latest Closed-Source Models
>
> We appreciate the reviewer’s valuable suggestion. During our experiments, GPT-4o and Gemini-1.5-Pro are commonly used proprietary models for comparison. As noted, these models are now relatively outdated as of late 2025. In response, we have added evaluations against the latest proprietary models, i.e., **GPT-5** and **Gemini-2.5-Pro**, in the revised version (ref. Table 1, Line 235-260). Here, we summarize the observations as follows:
>
> - On VideoMMMU and MMVU, our model lags behind GPT-5 and Gemini-2.5-Pro. Both benchmarks mainly consist of knowledge-intensive questions (e.g., Chemistry, Medicine, Engineering). In contrast, our approach focuses on improving the reasoning capability of the model via RL rather than injecting new knowledge during post-training. It is expected that models with richer scientific priors from pre-training outperform us on these science-heavy subsets.
>
> - On **Video-Holmes**, however, our method achieves **comparable performance** (42.7 vs. 45.0 vs. 46.7). Video-Holmes focuses on social reasoning, temporal causal inference, and contextual understanding rather than scientific facts. These results support that our method effectively enhances reasoning ability — the primary focus of our work.
>
> ---
>
> ### 2. Discussion on Generalization Beyond Video Reasoning
>
> We agree that broader evaluation is important. Although the method is designed for video reasoning, we observe **consistent cross-task and cross-scale generalization**:
>
> - **General video tasks:**
>   On Video-MME, performance improves from 59.3 (Video-R1 baseline [1]) to 60.3 using our method (ref. Table 1, Line 235-260).
> - **General image tasks:**
>   In Appendix B.3 (ref. Line 656-668), we show an average improvement of 4.7% over vanilla GRPO and 3.1% over Video-R1 across four image benchmarks.
> - **Smaller model setting (3B):**
>   In Appendix B.6 (ref. Line 730-746), we show that with 16-frame inputs, our method yields an average improvement of 1.7% on three video-reasoning benchmarks and 1.7% on two general video tasks.
>
> Overall, our approach improves VLLMs across video reasoning, general video tasks, and general image tasks, and across different model sizes, while adding only ~10% extra training time and requiring minimal implementation changes. This indicates that the method is not limited to a narrow domain but provides a **lightweight and broadly applicable enhancement** for modern RL post-training pipelines.
>
> We appreciate the reviewer’s valuable feedback.
>
> [1] Feng et al., Video-R1: Reinforcing Video Reasoning in MLLMs, NeurIPS 2025.

---

> > ### Comment · Reviewer_UDPQ · 2025-11-24
> >
> > Thanks a lot for the updates and candidness regarding the results. I do not have any more questions, but given my limited expertise, I will go through other reviewer discussions to make an informed decision. No change to score as of now.

---

### Official Review · Reviewer_DCwf · 2025-10-24

**Soundness:** 2
**Presentation:** 3
**Contribution:** 3
**Rating:** 4
**Confidence:** 4

**Summary:**

This paper proposes a way to select important tokens for reinforcement training visual LLMs to enhance their reasoning abilities. The method includes identifying visual-aware, temporal-aware and entropy-aware tokens (connecting words or logical words in my opinion) and only compute the loss based on those selected tokens.

**Strengths:**

1. The paper explores which tokens are more important to be optimized to enhance the reasoning process. This is a cheap and useful technique in general.

2. The paper is well-structured and clearly written, showing the background, proposed methods and experimental setup clearly.

3. The results seem promising as the token selection method yields better performance on reasoning-centric tasks such as video-Holmes. Ablation studies and various plots were made to support the findings.

**Weaknesses:**

1. I am a bit concerned regarding the theoretical foundation of this work. What is the actual contribution of the tokens that has actually been masked? For example, would these tokens introduce unwanted noise to the gradient w.r.t. the logits. A theoretical derivation by looking at the actual influence with or without the mask in the loss when taking the gradient w.r.t. the logits or even deeper in the network should be provided. __Moreover__, tokens that are visual-aware or temporal-aware, does not necessarily mean they need to be __optimized__ unless they really produces differences when sampling multiple paths. For example, I can have "cat" as a token that is visual-aware, but that cat appears in every samples I draw, and hence not producing useful information and can be excluded here. In my opinion, the tokens that should be optimized are the tokens that carry __information inconsistency__ across samples. I find this part of justification missing in the current manuscript.

2. I am also concerned about the entropy-aware tokens. I feel that the authors are trying to introduce the concept of information inconsistency here and to optimize tokens where the model is highly uncertain. However, I am not sure if entropy is a good metric, especially looking at the range of values it can get, a threshold might be difficult to draw, because different value on different tokens mean different things (even if you select via top-r% they are still inconsistent across reasoning paths). I find this part quite counter-intuitive and would leave this to other reviewers to decide.

Overall, I am not fully convinced by the proposed methodology, given that nowadays many factors may influence the model performance. Deeper analysis and theoretical justifications are needed for the current manuscript to be reasonably sound for publication.

**Questions:**

See weaknesses.

---

> ### Author Response · Authors · 2025-11-24
>
> We thank the reviewer for the thoughtful and constructive comments.
>
> ---
>
> ### **What is the actual contribution of the masked tokens?**
>
> Following your suggestion, we added a quantitative analysis in **Appendix B.2** (Line 605–653). By decomposing the final-layer gradient into contributions from *selected critical tokens* and *masked residual tokens*, we find that selected tokens yields larger and cleaner gradient updates than masked tokens. The mean gradient norm of critical tokens is 4.50 (about 3× that of masked tokens) and their cosine similarity with the full gradient averages 0.862 (vs. 0.572 for masked tokens). This shows that selected tokens dominate the effective update direction, while masked tokens contribute weak, less aligned gradients, indicating limited learning value and mainly adding perturbations. We further compare *training loss variance*. Video-KTR exhibits lower variance (0.4436 vs. 0.5528), confirming that masking these tokens reduces unnecessary fluctuations and stabilizes training.
>
> Qualitatively, our word-cloud and POS analyses further confirm that masked tokens are largely function words (e.g., determiners, auxiliary verbs, punctuation) with minimal semantic or visual-grounding value. These tokens mainly reflect syntactic continuation rather than reasoning-relevant content, explaining why excluding them benefits the model’s reasoning pathway.
>
> Together, these quantitative and qualitative results validate the effectiveness of our token-selection mechanism and highlight its role in guiding the model toward more meaningful and robust multimodal reasoning behavior.
>
> ---
>
> ### **Why do we optimize visual-aware or temporal-aware tokens?**
>
> Our aware-token mechanism directs updates to tokens that are essential for grounding the model’s reasoning in the visual and temporal content of the video. As shown in Table 2 (Line 297-306), updating only visual or only temporal tokens still yields clear gains over vanilla GRPO (+1.4 and +1.5 across benchmarks), confirming their importance.
>
> We agree that information inconsistency is an important signal of token value, and GRPO inherently leverages this property. However, optimizing for inconsistency is not the primary focus of our work. Our experiments show that visual and temporal awared tokens also carry substantial informational value, even when they may sometimes appear highly consistent across rollouts.
>
> In the case of “cat” mentioned in the context, this token may show low inconsistency and therefore contribute little from the perspective of information inconsistency. However, **from the standpoint of visual understanding**, it still carries key information about the visual content. The model should be encouraged to generate “cat” across different rollouts as this consistency is critical for accurate content understanding. By contrast, replacing “cat” with a token like “the” in the same context yields a deterministic, content-agnostic token that does not help interpret the video and is therefore not optimized.
>
> ---
>
> ### **Discussion on entropy-aware tokens**
>
> As the reviewer noted, entropy-aware tokens in our analysis are predominantly logical connectors such as “now,” “wait,” “seems,” which guide the model’s reasoning trajectory by marking transitions, expressing uncertainty, or prompting hypothesis refinement. Reinforcing these tokens when the reasoning path is correct helps the model solidify successful reasoning patterns.
>
> Recent studies have demonstrated that entropy-guided token selection is broadly effective. For instance, Wang et al. [1] highlight that high-entropy tokens dominate meaningful RL updates. Chen et al. [2] and Jiang et al. [3] further support entropy-guided optimization in reasoning models. Dang et al. [4] report similar patterns in video scenarios. However, these approaches are not modality-aware, while our work fulfills this gap by integrating visual-aware and temporal-aware token signals.
>
> ---
>
> ### **Discussion on threshold design**
>
> We agree that “different values on different tokens mean different things.” The top-r% strategy is indeed coarse, but it is simple yet empirically effective. As shown in Figure 7 (Line 376-386), we compared softmax, sigmoid, linear, and exponential mappings; top-r% achieved the best results, consistent with the Qwen's 80/20 strategy [1]. Designing a more principled thresholding method that accounts for differences in entropy semantics across reasoning paths remains an open research direction, and we leave this for future research.
>
> We appreciate the reviewer’s valuable feedback.
>
> [1] Wang et al., Beyond the 80/20 Rule: High-Entropy Minority Tokens Drive Effective Reinforcement Learning for LLM Reasoning, NeurIPS 2025.
>
> [2] Chen et al., Reasoning with Exploration: An Entropy Perspective, ArXiv 2025.
>
> [3] Jiang et al., Rethinking Entropy Regularization in Large Reasoning Models, ArXiv 2025.
>
> [4] Dang et al., Reinforcing Video Reasoning with Focused Thinking, ArXiv 2025.

---

> > ### Author Response · Authors · 2025-11-26
> >
> > Dear Reviewer DCwf,
> >
> > We hope that our responses have adequately addressed the concerns you raised in your review, and we would greatly appreciate your further feedback. Please let us know if there is anything further we can clarify.
> >
> > Best regards, The Authors

---

> ### Author Response · Authors · 2025-11-26
>
> Dear Reviewer DCwf,
>
> We hope that our responses have adequately addressed the concerns you raised in your review, and we would greatly appreciate your further feedback. Please let us know if there is anything further we can clarify.
>
> Best regards, The Authors

---

> > ### Comment · Reviewer_DCwf · 2025-11-26
> > **Response to the Authors**
> >
> > Thank you for the detailed explanation. I appreciate the authors efforst, but I still do not fully understand if the temporal or spatial tokens are fairly consistent across rollouts, why we still need to optimize them.
> >
> > The explanation to the entropy-aware tokens makes sense to me. I will increase my soundness score to reflect this, but I am sorry that the authors did not fully convince me about the necessity of other tokens, and hence I would maintain my overall rating.

---

> > > ### Author Response · Authors · 2025-11-27
> > >
> > > We thank the reviewer for the thoughtful comments.
> > >
> > > When temporal or visual tokens are described as “fairly consistent across rollouts,” this consistency can fall into two categories: (1) appearing at the same position across rollouts, and (2) appearing across rollouts but in different positions. Due to the inherent diversity introduced by VLLM rollouts, the first case almost never happens in practice. Therefore, our analysis focuses on the second case.
> > >
> > > ---
> > >
> > > ## 1. Same Tokens Can Carry Different Meanings Across Rollouts
> > >
> > > A single isolated token only serves as a pointer to an entity, action, or event, and its semantic contribution is entirely determined by its context. Even if a token appears in all rollouts, its role in the reasoning chain can be fundamentally different. Below is a specific example from the Video-Holmes benchmark:
> > >
> > > **[Question]**
> > > What is the correct order of the following events?
> > >
> > > ① The woman wipes the bloodstains.  ② The man lies on the ground motionless.
> > >
> > > ③ The woman picks up a hammer.  ④ The man sits up holding his head.
> > >
> > > ⑤ The woman hangs up the phone.
> > >
> > > **[Rollout 1 — Correct]**
> > > “… 4. *The man then sits up holding his head*…
> > > 5. *Finally, the woman hangs up the phone* …”
> > >
> > > **[Rollout 2 — Incorrect]**
> > > “… 4. *The woman hangs up the phone*…
> > > 5. *Finally, the man sits up holding his head*…”
> > >
> > > In this example, **“man”, “woman”, “sits up”, “hangs up the phone”** are all temporal/visual-aware tokens that are consistent across rollouts. However, the two rollouts place the same events in **opposite order**, leading to different reasoning outcomes. These “identical” tokens express different meanings under different contexts and guide the model toward completely different reasoning paths: **consistent occurrence does not imply consistent semantics**.
> > >
> > > An even more extreme illustration:
> > >
> > > **[Dummy Rollout 1]**
> > > The man drove to buy breakfast for the woman.
> > >
> > > **[Dummy Rollout 2]**
> > > The woman drove to buy breakfast for the man.
> > >
> > > Even though *man*, *woman*, and *drove* appear consistently across rollouts, the underlying facts represented by the two rollouts are entirely different.
> > >
> > > Therefore, **consistent tokens across rollouts still require optimization**, because their position, context, and temporal alignment are what determine whether the model’s reasoning path is correct.
> > >
> > > ---
> > >
> > > ## 2. Additional Ablation Study
> > >
> > > To further examine the necessary to optimize, we conducted an additional ablation study. All settings are kept identical to the main experiments (training with up to 16 frames, evaluating with 32 frames), The results are shown below:
> > >
> > > **Mask Consistent Visual&Temporal (V&T)**: additionally mask visual-aware and temporal-aware tokens that appear in all rollouts
> > >
> > > **Mask Consistent Entropy&Visual&Temporal (E&V&T)**: further mask entropy-aware tokens that also appear in all rollouts
> > >
> > > | Method                   | Video-Holmes | VideoMMMU | MMVU | Avg  |
> > > |--------------------------|--------------|-----------|------|------|
> > > | Vanilla GRPO             | 38.8         | 49.8      | 64.8 | 51.1 |
> > > | **Video-KTR**            | **41.6**     | **52.6**  | **65.9** | **53.4** |
> > > | Mask Consistent V&T      | 38.6         | 48.8      | 63.9 | 50.4 |
> > > | Mask Consistent E&V&T    | 37.7         | 48.2      | 63.1 | 49.7 |
> > >
> > > We observe a clear performance drop when masking visual/temporal tokens that consistently appear across rollouts (53.4 → 50.4), and an additional drop when also masking consistent entropy tokens (49.7). This indicates that although these tokens appear in all rollouts, **they still carry useful information and provide meaningful optimization signals.** Therefore, updating these “consistent visual/temporal-aware tokens” is necessary.
> > >
> > > ---
> > >
> > > ## 3. Considering the Characteristics of Video Reasoning Tasks
> > >
> > > In video reasoning, the model must understand who did what and in what order. Visual-aware and temporal-aware tokens correspond directly to these elements—e.g., “the man stands up,” “the woman picks up the hammer,” or “then she calls the police.” These tokens serve as anchors for the event timeline. Even if such tokens appear in every rollout, their positions may differ, which directly affects the predicted event order and can even flip the subject and object of an action. Moreover, GRPO only provides a final-answer reward, offering no supervision for intermediate steps. Without reinforcing these anchor tokens, the model may mix up actors or misplace events. Thus, even tokens that consistently appear across rollouts still need to be optimized so the model learns to use them in the correct context.
> > >
> > > ---
> > >
> > > We hope the above clarifications are helpful, and we look forward to any additional comments you may have.

---

### Official Review · Reviewer_DmeU · 2025-10-28

**Soundness:** 3
**Presentation:** 3
**Contribution:** 3
**Rating:** 6
**Confidence:** 3

**Summary:**

This paper focuses on video reasoning with multimodal large language models. Existing methods rely on coarse sequence level rewards or single-factor token selection for reinforcement learning (RL). This work Video-KTR is a policy shaping framework for token-level RL that only look at three types of tokens, which are visual-aware tokens (i.e., tokens closely associated with visual input like appear, show, etc.), temporal-aware tokens (i.e., tokens sensitive to the temporal structure of videos like finally, first, etc.), and entropy tokens (i.e.,  reasoning-critical tokens like however, wait, seem, now, etc.). Therefore, Video-KTR learns semantically informative, modality-sensitive, and filters low-value tokens, which contributing to the strong performance across five benchmarks.

**Strengths:**

1. The general idea is simple and insightful by letting the model reinforce on 3 types of tokens that are crucial to video reasoning, but effective that has performance gain over 5 video reasoning benchmarks.
2. Clear ablation study of all combination of 3 types of tokens to show clearly what type of token matters the most and how each type of tokens contribute to the performance gain.
3. 7 research questions are insightful. For examples, by using the same dataset and training recipe, they made sure that the comparisons are fair. There are linguistic insights like visual-aware tokens are mainly nouns, temporal aware tokens emphasize verbs and pronouns, and entropy-aware tokens has a higher share of adjectives. A general insight for large language models is that the log-probability differences is a reliable and efficient signal for tracking prediction-confidence shifts.
4. Writing and images are clear and easy to understand.

**Weaknesses:**

1. The performance gain over the Vanilla GRPO for three types of tokens, despite being higher, but it's a small increase. When enabled all three types, the average delta performance gain is just 2.4% for the average of three benchmark, while other ablations show even smaller differences. Therefore, the effectiveness of the Video-KTR is limited from the results.

**Questions:**

Is the Video-KTR model initialized from Qwen2.5-VL? It's better to make clear about the base model for implementation in the paper.

---

> ### Author Response · Authors · 2025-11-24
>
> We thank the reviewer for the thoughtful and constructive comments.
> ### 1. Discussion on Effectiveness
>
> Achieving consistent improvements on video reasoning benchmarks is inherently challenging, as these datasets are designed to be difficult and current post-training methods generally yield only modest gains. For example, the recent Video-R1 method [1], which is widely regarded as a strong post-training baseline, improves over vanilla GRPO by only **1.1%** average score across the three reasoning benchmarks. Our Video-KTR, which is based on the same cold-start SFT checkpoint as Video-R1, further outperforms Video-R1 with an additional **1.2%** average improvement. This corresponds to a **2.3%** total improvement over the GRPO baseline—approximately doubling the gain achieved by Video-R1.
>
> The characteristics of the benchmarks further illustrate the difficulty:
> - On **Video-Holmes**, even the latest proprietary model **Gemini 2.5 Pro** only achieves an accuracy of **45.0**, while our model reaches **42.7**—highlighting the intrinsic difficulty of the task and the effectiveness of our method.
> - **VideoMMMU** and **MMVU** mainly consist of **knowledge-intensive questions**, such as the science-related subcategories (Chemistry, Engineering, Medicine), which require massive domain-specific knowledge. In contrast, our approach focuses on improving the reasoning capability of the model via algorithmic design rather than injecting new knowledge during post-training. Under this constraint, we highlight that these **2.4%** average gains are meaningful and non-trivial.
>
> In addition, our method is both broadly applicable and cost-efficient. As shown in Appendix B.3, across four **image benchmarks**, our method improves over vanilla GRPO by **4.7%** on average and over Video-R1 by 3.1%. We also validate the approach on a **3B model** (Appendix B.6, ref. Line 730-745), where we observe an average **1.7%** improvement on video reasoning tasks and a **1.7%** gain on general video tasks. Considering that our method requires only ~10% additional training time and minimal implementation overhead, the performance gains are reliable and practically valuable. To the best of our knowledge, this makes our approach one of the strongest post-training methods currently available.
>
>
> ---
>
>
> ### 2. About the Initialized Model
> We thank the reviewer for pointing out this clarity issue. Our Video-KTR is initialized from Video-R1's cold start SFT checkpoint (which is further fine-tuned from Qwen2.5-VL-7B by Video-R1's author [1]) for fair comparison. We have clarified this in **Section 4.1 (Experimental Setup – Implementation)** (ref. Line 231-232) in the revised version.
>
> We appreciate the reviewer’s valuable feedback.
>
> [1] Feng et al., Video-R1: Reinforcing Video Reasoning in MLLMs, NeurIPS 2025.

---

> > ### Comment · Reviewer_DmeU · 2025-11-25
> >
> > Thanks for the clarifications. I remain the positive rating.

---

### Author Response · Authors · 2025-11-29
**Response Summary to All Reviewers**

We thank all reviewers for their thoughtful and constructive feedback, and we appreciate the comments highlighting the practical aspects of our method.

Below, we summarize the key clarifications, new experiments, and additional analyses incorporated into the revised manuscript.

---

### **Empirical Evidence for the Effectiveness of Token Selection**

In response to **Reviewer DCwf’s** questions regarding (1) the effectiveness of masking tokens, (2) the role of entropy-aware tokens, and (3) the necessity of updating tokens that appear consistently across rollouts, we provide the following clarifications supported by additional analyses and experiments.

First, we conducted a new experiment based on **gradient decomposition**, which directly verifies the **effectiveness of masking non-critical tokens.** Selected tokens produce ~3× larger gradient norms and much higher alignment with the full gradient, while masked tokens mostly contribute noisy, low-value updates. Training loss variance is also reduced, confirming that masking tokens stabilizes optimization and strengthens meaningful learning.

Second, regarding **entropy-aware tokens**, we introduced recent studies showing that high-entropy tokens have been widely validated as effective indicators for guiding RL optimization in reasoning models. However, these works do not consider modality-aware token selection.

Third, for updating tokens that appear in every rollout, we highlight that **consistent appearance does not imply consistent semantics**. For example, in Video-Holmes, two rollouts may contain the same tokens yet express reversed event orders (“the man sits up then the woman hangs up” vs. “the woman hangs up then the man sits up”). Our new ablation shows that masking these “consistent” visual/temporal tokens leads to clear performance drops (e.g., **53.4 → 50.4**), demonstrating that such tokens remain essential learning signals.

---

### **Comparisons with Latest Proprietary Models**

Following **Reviewer UDPQ’s** suggestion, we added performance comparisons against the latest closed-source models **GPT-5** and **Gemini-2.5-Pro**, which are now included in the revised manuscript (Section 4: Experiments).

- On **knowledge-intensive benchmarks** (VideoMMMU, MMVU), our model performs below these models—expected given the lack of scientific pre-training.
- On **Video-Holmes**, our model is competitive to both Gemini-2.5-Pro and GPT5 (**42.7 vs. 45.0 vs. 46.7**), showing strong gains in reasoning capability, which is the focus of our method.

---

### **Robust Cross-Benchmark Performance of Video-KTR**

VideoMMMU and MMVU are knowledge-intensive video benchmarks covering chemistry, engineering, and medicine, while Video-Holmes is a challenging causal-reasoning dataset on which even Gemini-2.5-Pro reaches only 45% accuracy. Despite the difficulty, Video-KTR achieves a 2.3% average improvement over vanilla GRPO across the three benchmarks and reaches **42.7% on Video-Holmes, close to state-of-the-art proprietary models**. Beyond video reasoning, it also shows strong generalization, yielding +4.7% gains on four image benchmarks and +1.7% on a Qwen-2.5-VL-3B setting, with only ~10% additional training cost. These results highlight Video-KTR as a lightweight, efficient, and broadly applicable token-level RL enhancement.

---

Besides, in response to **Reviewer DmeU’s** concern about unclear base-model initialization, we have explicitly clarified this in the revised manuscript (Section 4.1, Experimental Setup – Implementation).

---

### Meta-Review · Area_Chair_Yzx9 · 2026-01-07

**Summary:**

The submission introduces a modality-aware policy shaping framework for multimodal reasoning with video inputs. The key finding is that emprically for token-level RL, it is beneficial to focus on three types of tokens identified by the authors, namely visual-aware tokens, temporal-aware tokens, and entropy tokens.

The reviewers initially gave ratings of 664, and raised multiple important questions. Specifically, the relatively small increase over vanilla GRPO (DmeU), the conceptual and empirical benefits of visual-aware, temporal-aware, and entropy-aware tokens (DCwf), the comparison with more recent closed-source models (UDPQ), and the generalizability of the proposed framework beyond the focused application domain (UDPQ). Based on the responses from both the authors and the reviewers, the AC believes that the main concerns have been addressed by the authors during rebuttal, therefore the submission is ready to be accepted by ICLR.

**Reviewer Concerns:**

Reviewer DmeU had questions about the relative small improvement over vanilla GRPO. The authors responded by emphasizing the challenges of the benchmarks, both the reviewer and the AC believe the concern was addressed.

Reviewer DCwf had questions about the "theoretical foundation", and the role of different token types. The reviewer acknowledged that the concern on contribution of masked tokens and visual-aware and temporal-aware have been addressed, but had remaining concerns on why tokens consistent across rollouts need to be optimized. The authors subsequently provided intuitive and quantitive analysis in response, which the AC believes have addressed the concern.

Reviewer UDPQ acknowledged that they had no more questions after the rebuttal discussion.

**Reviewer Scores:**

The AC believes that all reviewers will recommend ratings of 6 after the rebuttal.

---

### Decision · Program_Chairs · 2026-01-26

Accept (Poster)